# Assessment of Vocal Fold Stiffness by Means of High-Speed Videolaryngoscopy with Laryngotopography in Prediction of Early Glottic Malignancy: Preliminary Report

**DOI:** 10.3390/cancers14194697

**Published:** 2022-09-27

**Authors:** Justyna Kaluza, Ewa Niebudek-Bogusz, Jakub Malinowski, Pawel Strumillo, Wioletta Pietruszewska

**Affiliations:** 1Institute of Electronics, Lodz University of Technology, 90-924 Lodz, Poland; 2Department of Otolaryngology, Head and Neck Oncology, Medical University of Lodz, 90-001 Lodz, Poland

**Keywords:** high-speed videolaryngoscopy, laryngeal cancer, laryngotopography

## Abstract

**Simple Summary:**

The method described in our manuscript can help to objectively assess the vibration of each vocal fold using larygotopographic analysis of high-speed videoendoscopy (HSV) recordings. We have developed image processing and analysis procedures to detect vocal fold regions in HSV films and quantitatively analyze their shape and kinematics. We proposed the term Stiffness Asymmetry Index which can provide valuable information on the texture and kinematic properties of individual vocal fold tissues, which can be important in the diagnosis of early glottis cancer. Our study showed that a low value of SAI indicated large, non-vibrating vocal fold areas, characteristic of infiltrative lesions such as invasive carcinoma. This important clinical information can help to assess the depth of vocal fold invasion before direct histologic examination and discriminate benign from malignant lesions.

**Abstract:**

One of the most important challenges in laryngological practice is the early diagnosis of laryngeal cancer. Detection of non-vibrating areas affected by neoplastic lesions of the vocal folds can be crucial in the recognition of early cancerogenous infiltration. Glottal pathologies associated with abnormal vibration patterns of the vocal folds can be detected and quantified using High-speed Videolaryngoscopy (HSV), also in subjects with severe voice disorders, and analyzed with the aid of computer image processing procedures. We present a method that enables the assessment of vocal fold pathologies with the use of HSV. The calculated laryngotopographic (LTG) maps of the vocal folds based on HSV allowed for a detailed characterization of vibration patterns and abnormalities in different regions of the vocal folds. We verified our methods with HSV recordings from 31 subjects with a normophonic voice and benign and malignant vocal fold lesions. We proposed the novel Stiffness Asymmetry Index (SAI) to differentiate between early glottis cancer (SAI = 0.65 ± 0.18) and benign vocal fold masses (SAI = 0.16 ± 0.13). Our results showed that these glottal pathologies might be noninvasively distinguished prior to histopathological examination. However, this needs to be confirmed by further research on larger groups of benign and malignant laryngeal lesions.

## 1. Introduction

Voice production is a very complicated function of the larynx, which involves interrelated physiologic, biomechanical, and aerodynamic mechanisms. Vocal fold vibrations play an essential role in this process [1]. The distribution of vocal fold oscillations affects the quality of voice. Therefore, accurate visualization of vocal fold vibrations is crucial in the diagnosis of various laryngeal disorders [2,3,4]. Currently, the most popular imaging tool for the assessment of phonatory movements of the vocal folds is laryngovideostroboscopy (LVS). It is important to note that stroboscopic imaging is not suitable for the reconstruction of aperiodic motion of the vocal folds, which occurs in many laryngeal pathologies. Additionally, during LVS examination, the subject needs to sustain phonation for a considerably long time, which is often impossible for patients with severe glottal pathologies.

In recent years high-speed videolaryngoscopy (HSV) has become an important method of visualizing vocal fold vibrations [5,6,7,8]. It can achieve image frame rates in the range of 2000 to 6000 frames per second (fps). Therefore, in HSV, the time needed to capture enough vibrations of the vocal folds is less than one second. In contrast to laryngovideostroboscopy, it allows for the assessment of aperiodic phonation and visualization of intracycle vibratory differences and can also be used with subjects whose phonation period is short due to severe voice disorders [9,10,11]. Furthermore, HSV provides for a more reliable analysis of glottal phonatory movements than VLS [4,12]. Imaging the larynx with high-speed videolaryngoscopy has facilitated the objective diagnosis of laryngeal pathology.

One of the most important challenges in laryngological practice is the early diagnosis of laryngeal cancer. Laryngeal carcinoma (LC) accounts for over one-third of all head and neck malignancies, which makes it one of the most common neoplasms in this anatomic region. Laryngeal squamous cell carcinomas historically account for around 85–90 percent of all LCs. Tobacco and alcohol are well-known risk factors, although cancer etiology is complicated, with both hereditary and environmental variables playing a role in the disease’s development. Despite the introduction of improved diagnostic and therapeutic methods, the 5-year survival rate has remained low for many years: 63.2% in 1975 and 60.4% in 2010. Premalignant laryngeal lesions (PLLs) precede approximately 90% of malignant tumors of the larynx [13,14,15]. They are described as structural changes in the mucosa caused by irritants or systemic diseases. PLLs exhibit many of the same symptoms as malignant laryngeal lesions, with the leading one being dysphonia which is most frequently manifested as hoarseness. Furthermore, the symptoms of benign glottal lesions (BGLs) are like those of early glottic carcinoma. Most benign larynx or glottal pathologies, e.g., polyps and edema, affect the vocal folds themselves, causing dysphonia. Therefore, early detection of symptoms typical for laryngeal cancer remains difficult.

As a result, efforts have continually been made to develop new endoscopic examination procedures. Toluidine blue and Lugol’s solution staining were among the first, although they resulted in a high rate of false positive diagnoses and required general anesthesia for direct laryngoscopy. In recent years, biological endoscopic evaluation has allowed for the visualization of neoangiogenesis and has made the larynx more accessible for investigation and monitoring of lesions. Narrow band imaging (NBI) and Storz SPIES are used to detect lesions smaller than 5mm as well as carcinomas in situ. Some studies, ours among them, have previously demonstrated the major utility of recently applied biological endoscope technologies, particularly NBI, in the diagnosis of oropharyngeal and laryngeal pathological lesions [16,17,18,19].

Recently, HSV has been recommended for the evaluation of organic vocal fold lesions because of the possibility of assessing the vocal fold vibratory function regardless of the regularity or irregularity of its behavior. Irregular vibrational cycles are characteristic of hypertrophic masses of the glottis, often occurring in an asymmetric form. Some authors emphasize that abnormalities in the vibratory function of the vocal folds may provide an accurate diagnosis in differentiating between benign and malignant glottal lesions (MGLs) [20,21]. Documenting the structural and dynamic characteristics of the vocal folds during phonation by means of HSV allows the vibrational movement to be observed in detail and can be useful in the diagnosis of suspected malignancy in glottal lesions [22]. This fact has important implications for the prognosis and choice of a method of treatment.

Many algorithms have been proposed for analyzing high-speed videolaryngoscopy images. The focus has been on developing algorithms for segmenting glottis images and determining geometric and time-related parameters of the glottal area [23,24,25]. In recent years, there have also been reports of using deep learning algorithms to identify the glottal area [26] as well as using optical flow algorithms [27]. Further, in our recent work [28], we have proposed a novel method for automatic segmentation of glottal images from HSV optimized by parameters derived from acoustic recordings during the phonation of the sustained vowel/i:/.

An original and effective approach to the analysis of laryngeal HSV images has been proposed by Japanese authors [22,29,30]. In their method, referred to as laryngotopography, a fast Fourier transform was used to detect the temporal variation in the brightness of different topographic regions in a sequence of laryngeal HSV images. This approach enabled the detection of periodic variations of the glottal area [31]. The fundamental frequency F0 and phase relations of these movements were determined.

However, to optimize the clinical use of the HSV technology in otolaryngological practice, it has become necessary to develop different techniques to analyze vocal parameters, quantitatively describing the vibratory behavior of the vocal folds and facilitating differentiation between organic glottal lesions. This is particularly important as, to date, no objective diagnostic tool has been made available that would allow for preliminary differentiation between benign vocal fold lesions and early glottic cancer prior to histopathological examination.

The aim of this study was to determine the differential coefficient between benign and malignant vocal fold lesions based on laryngotopographic images obtained in HSV examination.

## 2. Materials and Methods

### 2.1. Participants, Methods of the Examination, HSV Properties

Laryngeal HSV recordings were taken of 42 subjects hospitalized at the Department of Otolaryngology, Head and Neck Oncology of the Medical University of Lodz. A precondition for including an HSV recording in this study was that the entire vocal folds were correctly visualized, e.g., recordings where a tilting epiglottis obscured the anterior commissure or arytenoid cartilages blocked out the posterior part of the vocal folds were not included in the study. The study group consisted of 21 patients who had been diagnosed with vocal fold hypertrophy and suffered from voice disorders and 10 patients who were hospitalized due to other otolaryngological diseases (e.g., before septoplasty, myringoplasty, lateral bronchial cyst excision, etc.) but presented a normophonic voice and no vocal fold pathology (designated as the N subjects). Among the patients with voice disorders (the dysphonic group), 10 had primary benign glottal lesions, namely polyps (designated as the B subjects), whereas 11 had primary malignant or cancerous lesions of the vocal folds (designated as the M subjects). The final diagnosis was made based on histopathological examination of tissue specimens from the hypertrophic lesions, and in the malignant group, squamous cell carcinoma was found in all cases. Patients with early glottic cancer (T1N0M0) were included in the study. Patients with cancer involving, in addition to one vocal fold, the laryngeal ventricle, vestibular fold or anterior commissure, or anterior part of the contralateral vocal fold were excluded.

The dysphonic group included 10 men and 11 women (benign lesion—3 men, 7 women, malignant lesions—7 men, 4 women). Their ages ranged from 35 to 72 years, with a mean age of 55.2 years. The normophonic group consisted of 2 men and 8 women aged from 27 to 68 years, with a mean age of 47.1 years. The number of analyzed recordings is presented in Table 1.

All subjects underwent laryngological examination, followed by larynx imaging with a high-speed camera. HSV recordings were made using the Advanced Larynx Imager System (ALIS) (Diagnova Technologies, Wroclaw, Poland) equipped with a laser diode light source (ALIS Lum-MF1) and a high-speed camera (ALIS Cam HS-1). Laryngeal endoscopy was performed with a rigid endoscope Fiegert-Endotech 12.4/7.2 with 4.8 mm thick fiber optic cable. The HSV recordings were made during stable phonation of the sustained vowel/i:/ at a comfortable level of pitch and volume. The obtained recordings were kymographically analyzed with the aid of the Diagnova Technologies software DiagnoScope Specialist ver. 1.3 (Wroclaw, Poland) dedicated for ALIS with additional modules for short- and long-term kymographic analysis with parameterization. A detailed description of the apparatus and software used is provided in another publication [32]. After the initial automatic image stabilization, the software generated a kymographic cross-section of a given point along the glottis, from where the region containing its central axis was selected. The software identified the position of the edges of the vocal folds and plotted a glottal width waveform graph (GWW) that reflected instantaneous changes in the glottal width at different time points. Next, to enable precise quantitative analysis, the parameters describing the regularity of vocal fold vibrations in subsequent cycles were determined based on the GWW. That is how the fundamental frequency F0 was identified.

The HSV images were captured at a recording rate of 2400 frames per second (fps) for resolutions of 512 × 480 pixels, 480 × 400 pixels, and 512 × 448 pixels, and at a recording rate of 3200 fps for resolutions of 448 × 400 pixels and 368 × 448 pixels.

### 2.2. Software Tools

The high-level programming language Python version 3.7 and Spyder 4 environment were used to implement the developed image processing and analysis algorithms. This software environment included an extensive package of standard libraries. The main package used was the OpenCV (Open-Source Computer Vision Library), which is an open-source library containing several hundred computer vision algorithms for image processing, video analysis, and object detection [33]. The second main library was SciPy, which was used to perform operations on signals, e.g., computing Fourier transform spectra, B-spline interpolation, and eliminating offsets between the successive frames of recordings [34]. Other software libraries, such as NumPy [35] for creating and operating on image matrices and Matplotlib [36] for creating graphs and diagramming the results, were also used.

### 2.3. Image Pre-Processing

Before the main software algorithms were run to analyze the images of the larynx, image pre-processing procedures were performed to ensure that the images were of sufficiently good quality. Figure 1 illustrates the sequence of the image-preprocessing procedures implemented.

The blue component in the RGB color images was used to reduce glare caused by the light source (steps 2 and 3a). The red component, on the other hand, was used to enhance the difference in the level of brightness between the vocal folds and the glottal area (steps 2 and 3b). Next, image offsets reaching up to a dozen pixels between subsequent images were reduced (step 4). These offsets significantly affected the results of the analysis. They were reduced by correlating every two consecutive images and finding the coordinates for which the best fit between images occurred. The next step was to find in the HSV recordings the image with the largest glottic opening. Once found, the software user marked two points delineating the main axis of the glottal area (step 5) on the image selected. Finally, the inclination of the axis was used as the angle by which the image was rotated to align the axis vertically (step 6).

### 2.4. Processing and Analysis of HSV Images

After vertically aligning the glottal images and removing the offsets between consecutive images in the HSV recordings, further processing and analysis were conducted. Figure 2 shows the entire data processing process, from glottal images to the calculation of laryngotopographic parameters for the HSV recordings of the studied subject groups.

The first step was to identify the region of the vocal folds in the glottis image. This region was defined as the region of interest (ROI) and was the focus of further image analysis. To this end, a vertically aligned image of the larynx was displayed to the user. The user marked up to 20 evenly spaced points along the outer edges of the vocal folds (see image 2 in Figure 2). The points served as nodes for a third-order polynomial B-spline to be drawn around the vocal folds. The B-spline interpolation method maintains continuity between the curve segments connecting the nodes [37]. Figure 3a shows in more detail an example of the resulting B-spline curve located at the outer edges of the vocal folds as well as the identified areas of the left and the right vocal folds (Figure 3b).

Next, for each pixel coordinate in the image’s region surrounded by the B-spline curve, a signal was constructed representing changes in the brightness of this pixel in the analyzed HSV recording of the larynx (step 4 in Figure 2). The signal was called the brightness function and took values in the range of 0–255, i.e., over the entire brightness range of the image.

As can be seen in Figure 2, the brightness function was a periodic function for which the fundamental frequency *F*0 could be determined. For this purpose, we calculated a Fourier transform of this function. The Fourier amplitude spectrum is shown below the brightness function plot in Figure 2. Note the elevated maximum in the Fourier spectrum indicating the fundamental frequency *F0*. The procedure described above was performed for each pixel in the HSV recording of the larynx. On this basis, we could create laryngotopographic maps in which we used colors to label areas of the glottis showing regular periodic movements at the fundamental frequency *F*0. An example of the laryngotopographic map superimposed on the grayscale image of the glottis is shown in Figure 4a. Note the arrows coming from two different regions of the image. The blue arrow originates in a region where regular periodic motion was detected and reflected in the Fourier spectrum by a high harmonic located at *F*0 = 280 Hz (Figure 4b). This region has been colored. The green arrow, on the other hand, took its origin from a region where no regular periodic motion was detected, which was manifested by the absence of a harmonic at the fundamental frequency *F*0.

The following rule was adopted to color other regions and assign to them the regions characterized by brightness changes at a rate corresponding to the fundamental frequency.

The fundamental frequency amplitude map was built by reading the amplitude value of the harmonic corresponding to the fundamental frequency *F*0 in the Fourier amplitude spectrum for each pixel of the map. The amplitudes were then normalized by dividing all the values by the value of the highest amplitude in the whole image. Therefore, the range of amplitude values was scaled to the range between 0 and 1. Although these amplitudes took small values in some parts of the vocal folds (see, for example, the green graph in Figure 4b), it was also possible to determine the fundamental frequency *F*0 for the selected pixel map. However, following a consultation with a phoniatrician, a threshold value of 0.1 was chosen as a condition for assigning a color to a map point to indicate that that point exhibited clear harmonic oscillations. The amplitude laryngotopographic maps are shown in the fourth column of images in Figure.

The phase map, on the other hand, was constructed from the Fourier phase spectrum. It can be interpreted as a map of phase delays in the harmonic movements of the different regions of the vocal folds. The phase values are given in radians and range from -π to π. The phase laryngotopographic maps are shown in the fifth column of images in Figure. Different phase values are assigned different colors.

The anatomical areas of the glottis, which were identified as vibrating at the fundamental frequency, formed the basis for proposing an index to quantitatively describe synchronous vibrations of the left and right vocal folds. The aim was to determine the ratio of the number of pixels for which the fundamental frequency was detected in the area of each fold, expressed as the number of image pixels. The previously determined axis of symmetry and area of the vocal folds were used to obtain a split between the left and right vocal folds. The index, which we named the stiffness asymmetry index (SAI), is defined as follows:(1)$$\mathrm{SAI}=1-\frac{\;\;\;\;\;\;\frac{vibrating\;area\;in\;the\;affected\;vocal\;fold}{area\;of\;the\;affected\;vocal\;fold}\;\;\;\;\;\;}{\frac{vibrating\;area\;in\;the\;\mathrm{non}-\mathrm{affected}\;vocal\;fold}{area\;of\;the\;\mathrm{non}-\mathrm{affected}\;\;vocal\;fold}}=1-\frac{\;\;\;\;\;\frac{AVF_{F0}}{AVF}\;\;\;\;\;}{\frac{NAVF_{F0}}{NAVF}}$$
where

*AVF_F_*_0_—vibrating area in the affected vocal fold

*AVF*—area of the affected vocal fold

*NAVF_F_*_0_—vibrating area in the non-affected vocal fold

*NAVF*—area of the non-affected vocal fold

SAI can be used to compare affected and non-affected vocal folds. It was inspired by the observation that early-stage glottal malignant lesions usually occurred asymmetrically, involving either the left or right vocal fold. Figure 5 shows a diagram of the vocal folds characterized by different degrees of asymmetry in the areas of vibration of the left and the right vocal folds, indicated by the shaded areas. The corresponding SAI index values are shown below each diagrammed case. When this parameter was equal to 0, the vibratory motion of the vocal folds was symmetrical. The greater the value of this parameter, the greater the asymmetry in the vibration of the left and right vocal folds, e.g., a value of 0.75 means high asymmetry. Example graphs of laryngotopographic maps with different SAI values are shown in Figure 5.

## 3. Results

The sequence of image processing and analysis procedures, as described in Section 2 and summarized in Figure 2, was applied to a selection of 31 HSV recordings collected from 10 normophonic subjects and patients diagnosed with benign (10 subjects) or malignant (11 subjects) glottal lesions. Vocal fold stiffness (separately for the left and right vocal folds) was assessed for each group of subjects by identifying areas of tissue exhibiting synchronous vibration at the fundamental frequency *F*0. The index defined in Equation (1) is the measure we propose to compare asymmetries in the vibratory activity of the vocal folds.

The identified vibration frequencies (using laryngotopography) were compared with the results obtained from the kymographic analysis based on GWW (glottal width waveform) to verify whether the developed algorithm correctly calculated the fundamental frequency from the HSV recordings. The polar plots in Figure 6 show a comparison of the calculated fundamental vibration frequencies for each subject in the examined groups. The maximum difference observed was 20.6 Hz, and the average difference was 6.2 Hz. This was a fairly accurate result given the frequency resolution that can be achieved with the imaging methods compared. Note that the detected fundamental frequencies ranged from 90 Hz to 370 Hz.

Figure 7 shows the representative HSV images and laryngotopographic maps, i.e., the regions where the fundamental frequency was detected for each group of subjects. Note that for the normophonic case (Figure 7a), the distribution of the areas where the fundamental frequency was identified (indicated by the shaded areas) is close to the symmetry along the vertical axis. In benign lesions (Figure 7b), it is very common to have an area of regurgitation within the glottis. The unshaded region on the laryngotopographic map, which mainly overlapped with the region of polyps’ localization, indicated no detected vibrations at the fundamental frequency F0. Finally, in malignant lesions (Figure 7c), the area where *F*0 was detected significantly differed between the right and left vocal folds. In the presented case, the vibration was not present at all in the right vocal fold. This non-vibrating area, visualized in LTG, reflected a lack of visible mucosal wave—the wave-like movement of the vocal fold cover—in the right vocal fold due to tumor infiltration. The quantitative measure of these asymmetric vibration patterns was reflected in the variable values of SAI. Laryngotopographic maps constructed from LHSV recordings enabled visualization of vocal fold movements during phonatory cycles. The fundamental frequency maps indicated regions of the vocal folds that vibrated at the fundamental frequency *F*0 (the third column of images in Figure 7). The amplitude maps of the fundamental frequency represented the intensity of these vibrations (the fourth column of images in Figure 7).

Finally, the phase maps represented the phase shifts in the harmonic movements of the different regions of the vocal folds (the fifth column of images in Figure 7). These shifts are given in radians, and different palette colors are assigned to represent these phase shifts. Note that for a normophonic individual (Figure 7a), the phases did not significantly differ (similar palette colors are assigned to the vocal folds), indicating that the vocal folds vibrate in unison. On the contrary, for laryngeal lesions, we observed a different color in the phase maps indicating desynchronization of the vibration of the different regions of the vocal folds.

Table 2 summarizes the results of the analysis conducted and presents the parameter values for normophonic subjects and for subjects with benign and malignant lesions.

The box-and-whisker plot in Figure 8 shows the distribution of the SAI values for each group of subjects. Benign lesions were characterized by a wider range of values than normophonic cases, indicating that some types of lesions did not affect the regularity of fold vibrations, whereas, for others, substantial asymmetry in the vocal fold vibration patterns was detected. In contrast, for malignant lesions, the SAI values were clustered at higher values, exceeding 0.42, indicating significant asymmetry of vibration between vocal folds. Thus, malignant lesions can be clearly distinguished from normophonic cases and benign lesions based on SAI values.

### 3.1. Statistical Significance of the Results

A non-parametric Mann-Whitney test [39] was performed in the groups to test whether the differences in the parameter values were statistically significant: benign-normal, malignant-normal, and benign-malignant. Table 3 contains the *p*-values for the pairs of compared groups. The performed test showed a statistically significant difference in the SAI values between the group pairs: malignant-normal and malignant-normal lesions. On the other hand, the group of benign-normal lesions should be considered similar. For men (19) and women (12), the performed test also showed a statistically significant difference. The value of *p* is higher than for all subjects due to the small number of cases (see Table 3).

### 3.2. Calculation Time of Analysis

Program execution time was also studied. There were two parts to the program—the computational part and the interactive part. Measurements were made in 10 randomly selected data sets. The average computing time was 59.3 s, while the time for interaction was 50.9 s, which may be explained by the fact that the person performing the test knew the program well. For new users, the time of interaction would certainly be longer. In the interactive part, the user was asked to place approximately 20 points on the image of the larynx, indicating the outer edges of the vocal folds.

## 4. Discussion

Direct observation of the vocal fold oscillations and their objective assessment is important for making a proper diagnosis of various pathological changes of the glottis. With the availability of high-speed video (HSV) laryngoscopy, there has been interest in developing various techniques of analysis to quantitatively describe the vibratory behavior of the vocal folds in normophonic and dysphonic subjects. However, to optimize the clinical use of the HSV technology for otolaryngological practice, it has become necessary to find quantitative measures that would enable differentiation between benign and malignant vocal fold lesions. Several laryngeal HSV-based analysis methods have recently been developed [4,22,40,41], e.g., digital kymography, glottal area waveforms (GAW) analysis, phonovibrography, and laryngotopography (LTG). The most frequent way to quantify vocal fold oscillations in HSV is to analyze the changes in the glottal area GAW [24,42] or glottal width over time by plotting a glottal width waveform [43]. GAW analysis provides valuable preliminary information about glottal vibration characteristics, but it does not allow for quantitative lateral comparisons of the vibration patterns in the left and the right vocal folds [44], which is essential in the assessment of organic lesions of the glottis, because hypertrophic masses of glottis often occur in an asymmetrical form.

In this paper, we have proposed the use of laryngotopography as a method to detect periodic and non-periodic phonatory movements of the vocal folds in laryngeal HSV recordings performed on normophonic (control group) and dysphonic subjects with hypertrophic masses of the vocal folds (study group). In the study group, the final diagnosis was made based on a histopathological examination of the removed hypertrophic lesions. Among the dysphonic patients, benign vocal fold lesions (BVFL) were diagnosed in 10 subjects, whereas malignant or cancerous lesions of the vocal folds were found in 11 patients. We quantified and compared phonatory movements in the left and right vocal fold between the control group, the patients with voice disorders caused by benign glottic lesions, and those with voice disorders due to glottic cancer.

LTG involves a Fourier transformation of brightness for each pixel across a sequence of recorded HSV images to enable quantitative evaluation of the spatial characteristics of the amplitude and phase of vocal fold vibrations. The information about the amplitude and phase components at the fundamental frequency is of utmost importance. The distribution of maximum vibration amplitudes for different regions of the vocal folds is a valuable source of diagnostic data and provides important information about the structure of the vocal folds. It should be underlined that the membranous portion of the vocal folds that covers the vocal fold muscle is crucial in vocal fold vibration. It is a highly mobile structure with multiple layers and with special components (cells, proteins, and matrix scaffolding) that generate a wave-like motion across the superior surface of the vocal fold. It is called the mucosal wave and is only visible during VLS or HSV examination. Any changes in this structure alter the pliability of the vocal folds, impeding their ability to vibrate as they become ’stiffer’ [3,45]. The LTG analysis of HSV recordings enables visualization of the reduction in the vibratory amplitude in the stiffer (affected) vocal fold and compares it with the more pliable (non-affected) vocal fold.

The laryngotopographic maps are presented as images in which the spectral parameters, i.e., fundamental frequency, amplitudes, and phases of the vocal fold vibrations, are assigned specific pseudocolors to simplify their qualitative interpretation. Additionally, the laryngotopographic maps show vocal fold fragments that are not vibrating, reflecting a lack of vibration amplitude and mucosal wave in these regions [22,46]. The phase delay in the vibrating areas of the vocal folds is also used as a parameter to differentiate between different vocal fold diseases [22,46,47].

Our study has shown that irregular vibrational cycles are characteristic of hypertrophic masses of the glottis, causing their asymmetrical shape. This can be explained by the left-right asymmetry of the material property between the vocal folds with or without hypertrophic masses or by the glottal insufficiency resulting from the entrapment of pathological masses between the free edges of the vocal fold. In the perceptual rating of HSV recordings, patients with hypertrophic laryngeal pathologies showed greater asymmetry in amplitude, mucosal wave and phase, and poorer glottal closure than vocally healthy subjects. LPT analysis of normophonic subjects has indicated quasi-symmetric vocal fold vibrations by providing information about similar viscoelastic properties of both vocal folds. We proposed a stiffness asymmetry index, based on which we were able to distinguish malignant lesions of the vocal folds from benign lesions of the vocal folds

In the case of glottic cancer in the affected vocal fold, the absence of vibratory amplitude in the region of the infiltrative lesion was visualized in LTG as a non-vibrating area [22,46]. Other studies reported that cancerous vocal fold [21,45] demonstrated the absence of vibration in the total area or area of the vocal fold. In laryngotopography, the analysis indicated that the non-vibrating area was a significant parameter observed in glottic cancer [22,46]. In our study, we have quantitatively compared the vibrations between the affected vocal fold and the non-affected vocal fold by calculating the stiffness asymmetry index SAI. We have found that SAI significantly differed between the malignant group and the normophonic group (*p* = 0.0001) and took the average value for the M and N groups as 0.65 and 0.1, respectively. Moreover, the study indicated a significant difference in the average value of SAI between patients with benign and malignant lesions of the vocal folds (*p* = 0.0001). The average value of SAI for the benign group was 0.16. Thus, this index could differentiate between benign and malignant masses of the vocal folds, and it also confirmed that greater disturbances of phonatory oscillations occurred in cancerous vocal folds than in polypoid vocal folds. These results are supported by clinical observations that in glottic cancer, the loss of vibratory amplitude and mucosal wave (non-vibrating area) is caused by a deep invasion of the vocal fold structures [21,22,46]. Complete loss of vibratory amplitude in patients with glottic cancer usually indicates deep involvement of the vocal fold, including not only deep mucosal involvement but also the involvement of a ligament or even a muscle in the neoplastic process [3]. Thus, malignant lesions not only increase the masses of the cancerous vocal fold but also diminish its pliability, causing stiffness of the affected vocal fold.

In turn, the study has found no significant differences between the normophonic and benign groups. The results can be explained by the fact that SAI better reflects increased stiffness of a vocal fold than increased masses. Benign vocal fold lesions, frequently in a polypoid form, result in an increased mass of the vocal fold with edema of the superficial layer of the vocal fold mucosa but not in increased stiffness of the vocal fold cover. In polypoid corditis, the mass is frequently isodense with the surrounding layer. It frequently has a jellylike consistency and oscillates well [3]. This means an increase in mass but not an increase in the stiffness of the vocal fold structures. Therefore, in the group with benign glottal lesions, the average value of the coefficient SAI was significantly lower than in the group with a malignant glottal lesion.

The proposed index may differentiate between organic lesions that lead to increased stiffness of the affected vocal fold. The findings reported in the literature on organic and hypertrophic glottal lesions analyzed with HSV have emphasized that asymmetry of vocal fold oscillation may be due to asymmetry in the mass, tension, and elasticity or pliability of vocal folds. We have proposed a quantitative parameter that compares the mechanical properties and the symmetry of each vocal fold. Some authors have emphasized that in the case of voice disorders caused by massive glottal damage, the assessment of vibration characteristics, e.g., the fundamental frequency F0 (pitch) and intensity, should be done for each vocal fold separately [44,46,48]. Characterizing the vibration of the individual vocal folds with the application of SAI can provide valuable information about the texture of the tissue of each vocal fold, which is important in the diagnosis of early glottic cancer. High values of SAI indicate large non-vibrating vocal fold areas characteristic of infiltrative lesions such as invasive carcinoma. This important clinical information can help assess the depth of the vocal fold invasion prior to a direct histologic examination and help guide otolaryngologists attempting endoscopic resection by cordectomy. The usefulness of an objective assessment of the stiffness of the vocal folds resulting from a deep infiltration of its structures in early glottic cancer for the surgeon prior to the surgery could not be overestimated.

The methodology detailed in this article for determining SAI is relatively time-consuming and requires appropriate training, especially if it is to be performed by a physician. The determination of SAI currently limits its use in daily clinical practice during the examination of an individual patient. The steepness of the learning curve remains to be verified in daily clinical practice. Furthermore, especially from an oncological point of view, an unequivocal statement on whether the application of SAI allows for differentiation between benign and neoplastic lesions in each vocal fold requires further study on a larger group of patients. Another limitation of this research was that only patients with unilateral lesions were included in the study group. We intentionally did this to determine the stiffness coefficient for the altered vocal fold and make a comparative analysis with a second healthy vocal fold in the same patient. During further studies, we plan to include patients with bilateral vocal fold lesions or patients with laryngeal cancer involving, in addition to the one vocal fold, the anterior commissure or anterior part of the contralateral vocal fold to test the clinical usefulness of the stiffness coefficient assessed before surgery.

## 5. Conclusions

In this study, we have shown that HSV recordings supported by computer image analysis techniques can offer important insight for otolaryngologists and phoniatricians in diagnosing glottal pathologies. Our findings can be summarized as follows: LTG analysis of HSV recording enabled the identification of abnormal regions of vibration in each vocal fold separately, which is important in the diagnosis of organic glottal lesions.The proposed stiffness asymmetry index (SAI) allowed for a quantitative comparison of vibrations between affected and non-affected vocal folds.LTG maps provided valuable quantitative information about vibrating regions of the vocal folds in terms of the intensity of the vibrations at the fundamental frequency F0 and the phase of these harmonics. Furthermore, it allowed for the detection of non-vibrating areas affected by neoplastic lesions of the vocal folds.The results obtained from the studied HSV recordings indicated that SAI has a significantly higher value in malignant vocal fold lesions than in benign vocal fold lesions and that it facilitated preliminary differentiation between early glottis cancer and benign vocal fold masses prior to histopathological examination.


Further work will focus on improving the graphical user interface of the developed computer program to make it more user-friendly for otolaryngologists and phoniatricians and to enable more efficient analysis of successively collected new HSV recordings.

## Figures and Tables

**Figure 1 cancers-14-04697-f001:**
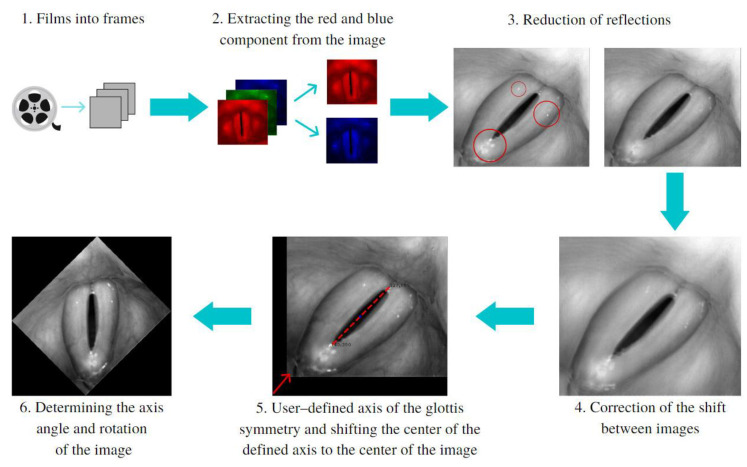
The multistep pre-processing of the LHSV images. Red circles (in step 3) indicate areas of the image reflections and the dashed line (in step 5) marks axis of the glottis.

**Figure 2 cancers-14-04697-f002:**
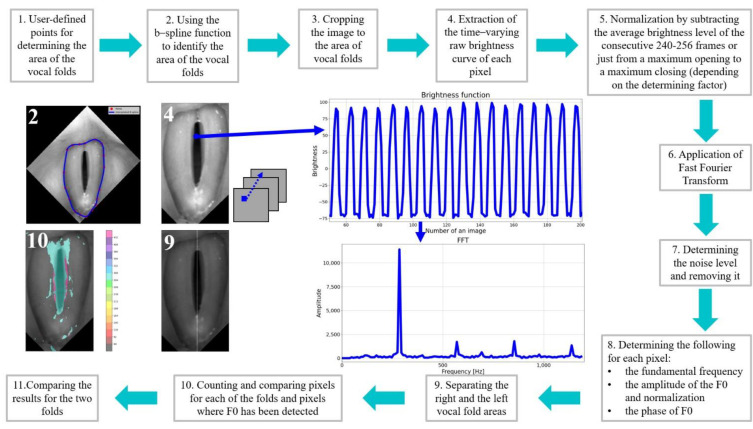
Sequence of procedures for processing and analysis of LHSV recordings.

**Figure 3 cancers-14-04697-f003:**
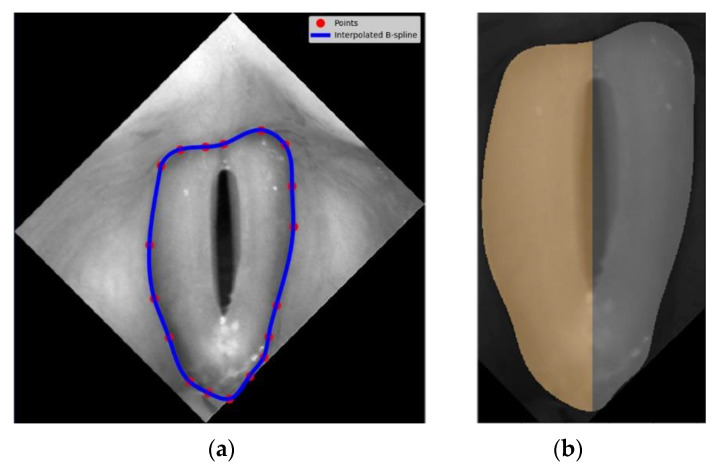
Identification of the region of interest for further image analysis procedures: (**a**) vocal fold area delineated with b-spline curves represented by blue line and the user-defined interpolation nodes, (**b**) identification of the left (grey shaded) and the right (orange shaded) vocal folds.

**Figure 4 cancers-14-04697-f004:**
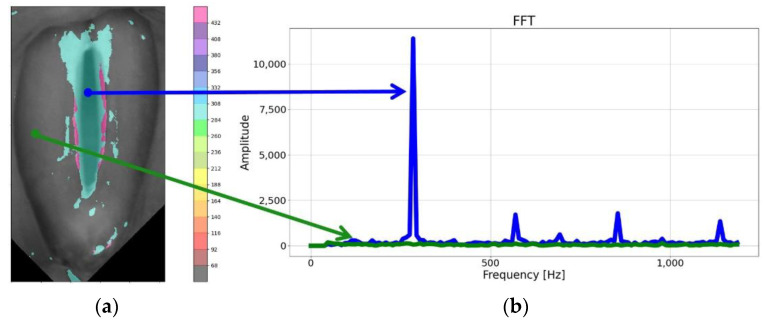
The laryngotopography of the fundamental frequency (**a**) and FFT plot (**b**) for two marked points of the LHSV image, the blue plot corresponds to the area of vibration of the vocal folds and the green plot corresponds to the area in which no regular periodic movement was detected. The color on the map was taken from a color palette (shown on the right-hand side of the image) in which different colors were assigned to different frequency ranges.

**Figure 5 cancers-14-04697-f005:**
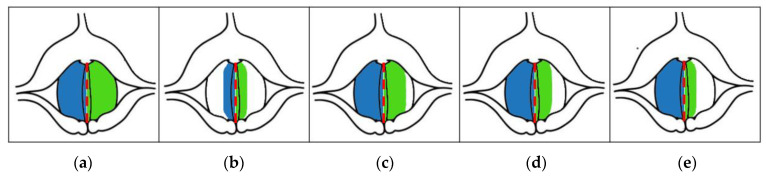
Illustration of the detected different size regions of the vocal folds vibrating at the fundamental frequency indicated by the different values of SAI and the corresponding values of the parameter (approximate values), (**a**) SAI = 0.0, (**b**) SAI = 0.0, (**c**) SAI = 0.25, (**d**) SAI = 0.5, (**e**) SAI = 0.75 [38]. The red dashed line marks the axis of symmetry of the glottis, and the blue and green areas correspond to the detected areas of vibration of the right and left vocal folds, respectively.

**Figure 6 cancers-14-04697-f006:**
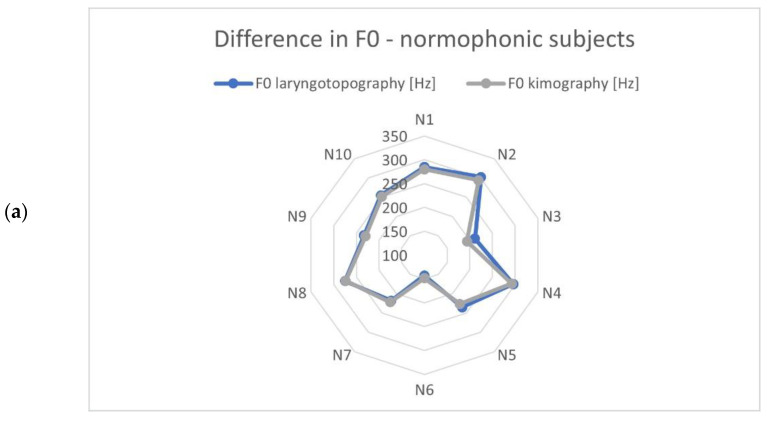
Comparison of the fundamental frequencies obtained from HSV recordings by means of laryngotopography and kymography analyses for each study group: (**a**) normophonic, (**b**) benign, and (**c**) malignant lesions. The scale of the polar diagrams is given in Hz and the codes N1-N10, B-10 and M1-M11 correspond to the normophonic subjects and subjects with benign and malignant lesions, respectively.

**Figure 7 cancers-14-04697-f007:**
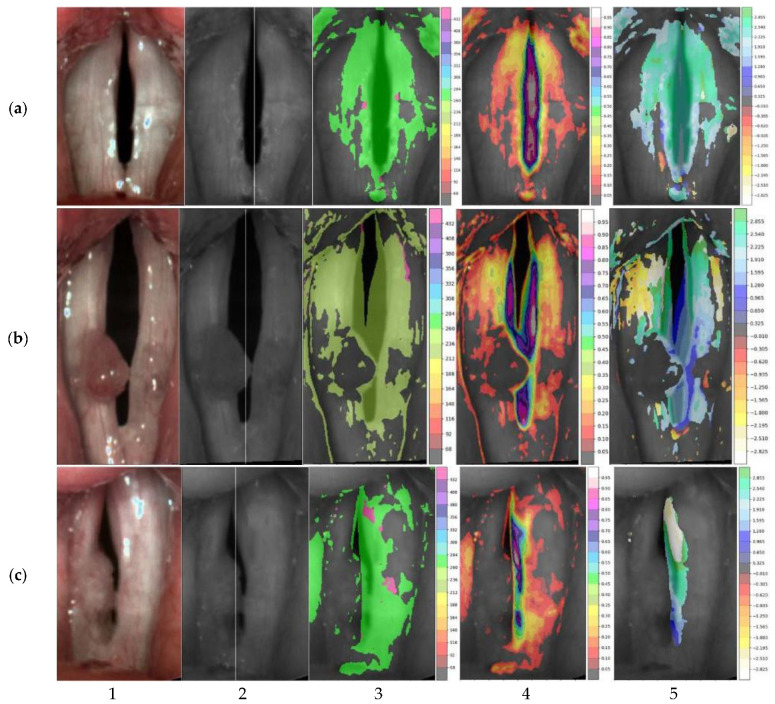
Representative HSV images recorded in normophonic subjects (**a**) and patients diagnosed with benign (**b**) and malignant glottal lesions (**c**). The first column shows the registered LHSV images, the second column shows their monochrome versions with the axis of symmetry drawn, the third column shows the amplitude maps of the fundamental frequency (with F0 values on colorbar), and the fourth and fifth columns show maps of the intensity (with its value on colorbar) of these harmonic vibrations and their phases correspondingly (with angle value in radians on colorbar). (**a**) SAI = 0.03. (**b**) SAI = 0.21. (**c**) SAI = 0.84.

**Figure 8 cancers-14-04697-f008:**
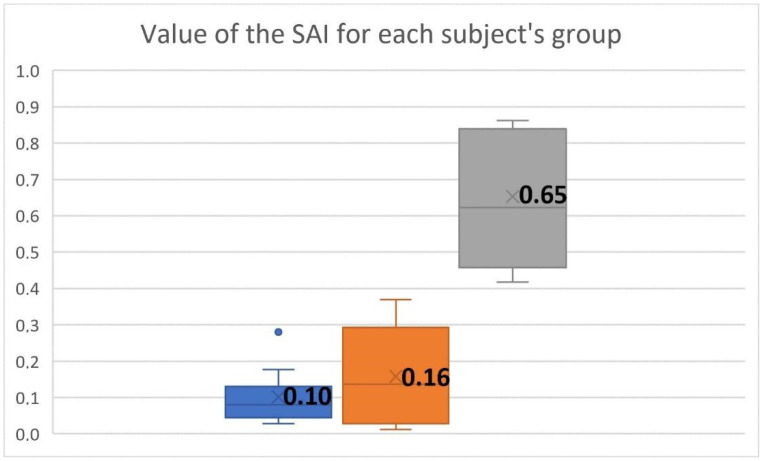
Box-and-whisker plots of the SAI values for each subject group, norms (blue box), benign (orange box), and malignant (gray box). The blue dot corresponds to an elevated SAI obtained for a single subject with a normophonic voice.

**Table 1 cancers-14-04697-t001:** Analyzed recordings.

Diagnosis	Number of HSV Recordings from Subjects	Number of Recordings Selected for Analysis
Normophonic	13	10
Benign glottal lesions	15	10
Malignant glottal lesions	14	11

**Table 2 cancers-14-04697-t002:** Values of SAI for each subject group.

Diagnosis	Number of Subjects	Average SAI
Normophonic	10	0.10 ± 0.07 (0.03–0.28)
Benign glottal lesions	10	0.16 ± 0.13 (0.01–0.37)
Malignant glottal lesions	11	0.65 ± 0.18 (0.42–0.86)

**Table 3 cancers-14-04697-t003:** *p*-values calculated for the Mann-Whitney statistical test to compare groups of subjects.

	Benign-Normal	Malignant-Normal	Benign-Malignant
*p*-value	0.7624	0.0001	0.0001
*p*-value men	0.5637	0.0404	0.0167
*p*-value women	0.4874	0.0066	0.0081

## Data Availability

Not applicable.

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
