# Peer review of "Assessment of Vocal Fold Stiffness by Means of High-Speed Videolaryngoscopy with Laryngotopography in Prediction of Early Glottic Malignancy: Preliminary Report"

_cancers, 2022, doi:10.3390/cancers14194697_

Round 1

Reviewer 1 Report

  • Would like to see a comparison between the results of the presented algorithm, with results of other algorithms tested by other papers/authors.
  • Column 5 in figure 7 had no clear explanation as to what it was presenting, as compared to the other columns in the figure. Further explain the images in column 5 such as the explanations presented for the other columns in the same figure.
  • Clearly state the exact name of the software used to obtain the kymographic cross section. 
  • Lack of statistical data was presented. Patients in the paper were not distinguished relative to sex (male/female). This is an important distinguishment to make as the fundamental frequency at which females phonate differs from that of male phonation. Further statistical comparisons should be presented between the males of the 3 groups, and the same between the females.

Author Response

Dear Reviewer,

Thank you very much for your effort and time taken to review our manuscript. We agree with all your comments and suggestions, and we have implemented all the suggested changes in the manuscript, below is the point-by-point reply to your comments:

  • Would like to see a comparison between the results of the presented algorithm, with results of other algorithms tested by other papers/authors.

Response: We have proposed a new parameter here and at the moment we have not found similar studies in the available literature to discuss. Although the literature on the use of HSV itself is quite extensive, a similar parameter to the SAI we have used has not yet been described in the literature.

  • Column 5 in figure 7 had no clear explanation as to what it was presenting, as compared to the other columns in the figure. Further explain the images in column 5 such as the explanations presented for the other columns in the same figure.

Response: Thank you for your attention. The section above graphic 7 has been added.

  • Clearly state the exact name of the software used to obtain the kymographic cross section. 

Response: We have added the name of the software starting with the line 152.

Added to the manuscript: The obtained recordings were kimographically analyzed with the aid of Diagnova Technologies software DiagnoScope Specialist ver. 1.3 dedicated for ALIS (Advanced Larynx Imager System) with additional modules for short-term and long-term kimographic analysis with parameterization.

  • Lack of statistical data was presented. Patients in the paper were not distinguished relative to sex (male/female). This is an important distinguishment to make as the fundamental frequency at which females phonate differs from that of male phonation. Further statistical comparisons should be presented between the males of the 3 groups, and the same between the females.

Response: We performed statistical tests separately for men and women and refined the number of patients by men and women. We have added this information in the manuscript starting at line 142 and line 381 , also in table 3.

Added to the manuscript: For male (19) and female (12) individuals, the performed test also showed statistically significant difference. The value of p is higher then for all subjects due to small number of cases (see Table 3).

Reviewer 2 Report

The authors presented a method that enables the assessment of vocal folds pathologies with the use of HSV in this manuscript. Overall, this study was well-designed and presented with ample data. However, some analyses are missing and shall be further provided to further improve the quality of this study.

The sensitivity, specificity, accuracy, positive and negative predictive values (PPV and NPV) of functional glottal assessment during LVS for differentiating malignant lesions should be provided.

Receiver operating characteristic (ROC) curves to decide on the cut-off level of LVS scoring for optimal sensitivity and specificity.

Most figures are not well presented and labeled with necessary captions. Authors must address them to meet the criteria from the journal.

Author Response

Reviewer 2.

Thank you very much for your thoughtful comments and efforts toward improving our manuscript. We would like to address your concerns and comments point by point:

  • The authors presented a method that enables the assessment of vocal folds pathologies with the use of HSV in this manuscript. Overall, this study was well-designed and presented with ample data. However, some analyses are missing and shall be further provided to further improve the quality of this study. The sensitivity, specificity, accuracy, positive and negative predictive values (PPV and NPV) of functional glottal assessment during LVS for differentiating malignant lesions should be provided. Receiver operating characteristic (ROC) curves to decide on the cut-off level of LVS scoring for optimal sensitivity and specificity.

Response: Thank you for your comments, we have carried out the analyses. 

The box-and-whisker plot (Figure 8) clearly shows that the extreme values for both the group of normophonic and benign subjects do not coincide with the extreme values of subjects with malignant lesions. On this basis, it can be concluded that there is a certain threshold that will unambiguously allow to classify malignant lesions. To check the thesis, we plotted a ROC curve for all patients and separately for the females and males. In all cases, the curve looks similar and passes throught the point (0,1), which confirms that there is a value of the SAI that unequivocally classifies subjects with malignant lesions from benign and normophonic.

In order to check whether we are able to determine the value of the parameter, which is able to clearly distinguish malignant lesions from benign and normophonic subjects, the ROC curve was drawn and the following statistical parameters were calculated: the accuracy, sensitivity, specificity, positive and negative predictive values, AUC and Youden index [40]. The results are presented in the Table below. However, the results need to be confirmed in larger groups of patients, so we are considering posting them as Supplementary Table S1.    

Table 4. The accuracy, sensitivity, specificity, positive and negative predictive values, ROC curve parameters: AUC and Youden index in differentiation between the groups.  

Subject’s group

Proposed cut-off point

AUC

Youden’s index

Accuracy

%

Sensitivity

%

Specificity

%

PPV

%

NPV

%

All subjects

0.40

1.0

1.0

100

100

100

100

100

Males

0.35

1.0

1.0

100

100

100

100

100

0.40

1.0

1.0

100

100

100

100

100

Females

0.40

1.0

1.0

100

100

100

100

100

0.45

1.0

1.0

100

100

100

100

100

  • Most figures are not well presented and labeled with necessary captions. Authors must address them to meet the criteria from the journal.

Response: Thank you for this suggestion, we have corrected the descriptions of Figures 1, 2 4, 5, 6, 7, 8 to correspond more closely to the content, making them more readable.